# FEDERATED LEARNING OF LARGE MODELS AT THE EDGE VIA PRINCIPAL SUB-MODEL TRAINING

## ABSTRACT

Limited compute, memory, and communication capabilities of edge users create a significant bottleneck for federated learning (FL) of large models. Current literature typically tackles the challenge with a heterogeneous client setting or allows training to be offloaded to the server. However, the former requires a fraction of clients to train near-full models, which may not be achievable at the edge; while the latter can compromise privacy with sharing of intermediate representations or labels. In this work, we consider a realistic, but much less explored, cross-device FL setting in which no client has the capacity to train a full large model nor is willing to share any intermediate representations with the server. To this end, we present Principal Sub-Model (PriSM) training methodology, which leverages models' low-rank structure and kernel orthogonality to train sub-models in the orthogonal kernel space. More specifically, by applying singular value decomposition to original kernels in the server model, PriSM first obtains a set of principal orthogonal kernels with importance weighed by their singular values. Thereafter, PriSM utilizes a novel sampling strategy that selects different subsets of the principal kernels independently to create sub-models for clients with reduced computation and communication requirements. Importantly, a kernel with a large singular value is assigned with a high sampling probability. Thus, each sub-model is a low-rank approximation of the full large model, and all clients together achieve nearly full coverage of the principal kernels. To further improve memory efficiency, PriSM exploits low-rank structure in intermediate representations and allows each sub-model to learn only a subset of them while still preserving training performance. Our extensive evaluations on multiple datasets in various resource-constrained settings demonstrate that PriSM can yield an improved performance of up to 10% compared to existing alternatives, when training sub-models with only 20% principal kernels ($\sim 5\%$ of the full server model.).

## 1 INTRODUCTION

Federated Learning (FL) is emerging as a popular paradigm for distributed and privacy-preserving machine learning as it allows local clients to perform ML optimization jointly without directly sharing local data (McMahan et al., 2017; Kairouz et al., 2021). Thus, it enables privacy protection on local data, and leverages distributed local training to attain a better global model. This creates opportunities for many edge devices rich in data to participate in the joint training without direct data sharing. For example, resource-limited smart home devices can train local vision or language models using private data, and achieve a server model that generalizes well to all users via FL (Pichai, 2019).

Despite significant progress in FL in the recent past, several crucial challenges still remain when moving to the edge. In particular, limited computation, memory, and communication capacities prevent clients from learning large models for leveraging vast amounts of local data at the clients. This problem is getting increasing attention in current literature (Diao et al., 2021; Horvath et al., 2021; Yao et al., 2021; Vepakomma et al., 2018; He et al., 2020). For example, recent works propose a sub-model training methodology by assigning clients with different subsets of server model depending on their available resources (Diao et al., 2021; Horvath et al., 2021; Yao et al., 2021). However, these works have an underlying assumption that some of the clients have sufficient resources to train a nearly full large model. In particular, methods like FedHM (Yao et al., 2021) that adapt low-rank compression to FL incur more memory footprint for intermediate representations, even for small

sub-models. As a result, server model size is limited by the clients with maximum computation, memory, and communication capacities. To overcome resource constraints on clients, other works (Vepakomma et al., 2018; He et al., 2020) change the training paradigm by splitting a model onto server and clients. The computational burden on the clients is therefore relieved as the dominant part of the burden is offloaded to the server. However, such a methodology requires sharing of intermediate representations and/or labels with the server, which directly leaks input information and potentially compromises privacy promises of FL.

Unlike prior works, this work targets an even more constrained and realistic setting at the edge, in which no client is capable of training a large model nor is willing to share any intermediate data and/or labels with the server. To this end, we propose Principal Sub-Model (PriSM) training, which at a high level, *allows each client to only train a small sub-model, while still enabling the server model to achieve comparable accuracy as the full-model training*.

The cornerstone of PriSM is the models' inherent low-rank structure, which is commonly used in reducing compute costs (Khodak et al., 2021; Denton et al., 2014). However, naive low-rank approximation in FL (Yao et al., 2021), where all clients only train top-$k$ kernels, incurs a notable accuracy drop, especially in very constrained settings. In Figure 1, we delve into the matter by showing the number of principal kernels required in the orthogonal space to accurately approximate each convolution layer in the first two *ResBlocks* in ResNet-18 (He et al., 2016) during FL training[1]. We observe that even at the end of the FL training, around half of the principal kernels are still needed to sufficiently approximate each convolution layer. We have similar findings for the remaining convolution layers (See Sec 4.3). Therefore, to avoid the reduction in server model capacity, it is essential to ensure that all server-side principal kernels are collaboratively trained on clients, especially when each client can only train a very small sub-model (e.g., $< 50\%$ of the server model).

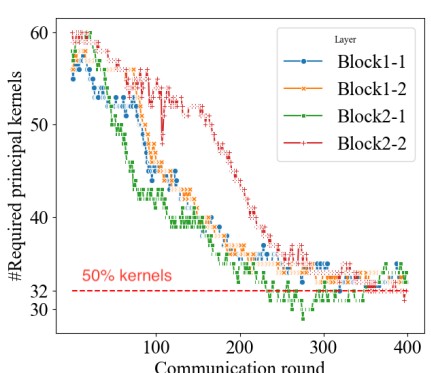

Figure 1: Number of principal kernels in the orthogonal space required to accurately approximate each of the two convolution layers in the first two ResBlocks in ResNet-18 during FL training. Block$i$-$j$ indicates $j$-th convolution layer in $i$-th ResBlock. Each of these convolution layers has 64 kernels.

Based on the above observations, PriSM employs a novel probabilistic strategy to select a subset of kernels and create a sub-model for each client as shown in Figure 2. More specifically, PriSM first converts the model into orthogonal space where original convolution kernels are decomposed into principal kernels using singular value decomposition (SVD). To approximate the original server model, PriSM utilizes a novel sampling process, where a principal kernel with a larger singular value has a higher sampling probability. The probabilistic process ensures that all sub-models can together provide nearly full coverage of the principal kernels, thus reaching the near full-model training performance with reduced costs on local computation and communication during sub-model aggregation. PriSM further improves memory efficiency by exploiting low-rank structure in intermediate activations and allows each client to learn only a subset of these representations while still preserving training performance. Thus, computation, memory, and communication bottlenecks at the edge are effectively resolved.

We conduct extensive evaluations for PriSM on vision and language tasks under resourced-constrained settings where no client is capable of training the large full model. In particular, we consider both resource constraints and heterogeneity in system capacities as well as data distribution. Our results demonstrate that PriSM delivers consistently better performance compared to other prior works, especially when participating clients have very limited capacities. For instance, on ResNet-18/CIFAR-10, we show that PriSM only incurs around $2\%$ and $3\%$ accuracy drop for i.i.d and highly non-i.i.d datasets under a very constrained setting where all clients train sub-models with only $20\%$ of the principal kernels, accounting for $\sim 5\%$ of the full server model. Compared to other solutions, PriSM improves the accuracy by up to $10\%$. Furthermore, we provide detailed insights into the performance gains attained by PriSM via 1) analyzing server model's rank structure during training; 2) profiling the kernel sampling process; 3) breaking down costs in the system.

---

[1]See Sec 4.3 for further details, especially for calculating the required number of principal kernels.

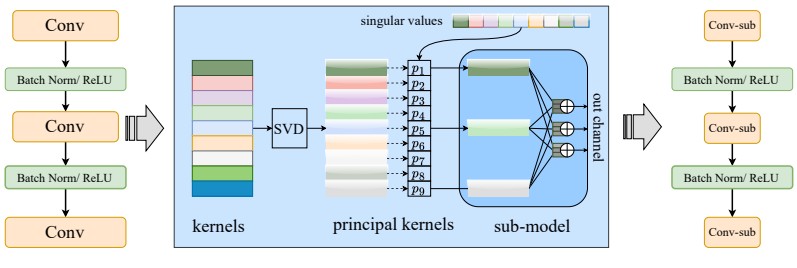

Figure 2: Creating clients' sub-models. PriSM samples a subset of principal kernels to create a client's sub-model that enjoys less computation and communication overhead. The singular value based sampling scheme ensures every sub-model approximates the full large model, and all sub-models together provide nearly full coverage of the principal kernels. PriSM further improves memory efficiency by allowing each sub-model to learn only a subset of intermediate representations.

## 2 RELATED WORKS

**Factorized Models**. Training neural networks with layer factorization has been extensively studied in prior literature (Denton et al., 2014; Khodak et al., 2021; Jaderberg et al., 2014; Novikov et al., 2015; Ioannou et al., 2016). Specifically, these works are based on the observation that well-trained neural networks have inherently low-rank structure and exhibit large-correlations across kernels. Hence, one can down-size the model with a low-rank approximation to provide significant reduction in computations thus speeding up training. Furthermore, this can make model training more affordable for resource-constrained devices. In addition to models' low-rank structure, works such as (Niu et al., 2022) also exploit low-rank structure in intermediate representations to reduce computation overhead.

**Resource-Constrained Federated Learning**. While federated learning opens the door for collaborative model training over edge users having rich (but private) data, the computation, memory, and communication footprint prohibits training of large models at the resource-constrained clients. To address these resource limitations in federated learning, a number of works have been proposed in the literature (Diao et al., 2021; Horvath et al., 2021; Yao et al., 2021; Diao et al., 2021; Horvath et al., 2021; Yao et al., 2021; Vepakomma et al., 2018; Poirot et al., 2019; Chopra et al., 2021; He et al., 2020). Particularly, in split learning (Vepakomma et al., 2018; Poirot et al., 2019; Chopra et al., 2021), the model is partitioned into two parts, one (small) part is assigned to clients for local training, while the other (large) part is outsourced to the server. He et al. (2020) proposes FedGKT that combines the model splitting approach with a bi-directional knowledge transfer technique between server and clients to achieve resource-constrained FL with much fewer communications than split learning. However, works such as split learning and FedGKT require sharing of intermediate activations (and in many cases, logits as well as labels) with the server, directly leaking input information and potentially compromising privacy promises of FL (Zhang et al., 2020).

The works closely related to ours are HeteroFL (Diao et al., 2021), FjORD (Horvath et al., 2021) and FedHM (Yao et al., 2021), that aim to enable participation of a resource-constrained client by letting it train a smaller sub-model based on its capabilities. In particular, HeteroFL and FjORD create sub-models for clients by selecting certain fixed number of original kernels of the server model. On the other hand, FedHM creates sub-models using fixed subsets of factorized principal kernels. However, in these works, the size of the server model gets limited by the clients with maximum computation, memory, and communication capacities, sacrificing the model performance. In particular, methods like FedHM incur more memory footprint for intermediate representations, even for small sub-models. This becomes even more critical in the realistic, cross-device FL setting wherein no client has the capacity to train a large model. While another work, FedPara (Hyeon-Woo et al., 2021), proposes a low-rank factorized model training to reduce communication costs, computational footprint still remains prohibitive as every client is required to perform full-model training.

Therefore, further efforts are still needed to effectively address the computation, memory, and communication bottleneck at the edge, while still preserving the privacy promises of FL.

## 3 METHOD

In this section, we first motivate our proposal, Principal random Sub-Model training (PriSM), with an observation of orthogonality in convolution layers. Then, we describe the details of PriSM.

**Notations–** $\|\cdot\|_F$: Frobenius norm. $\sigma_i$: $i$-th singular value in a matrix. $\circledast$: convolution. $\cdot$: matrix multiplication. $\langle \cdot, \cdot \rangle$: sum of element-wise multiplication or inner product. $tr(A)$: trace of a matrix.

### 3.1 MOTIVATION: AN OBSERVATION ON ORTHOGONALITY

We consider a convolution layer with kernels $W \in \mathbb{R}^{N \times M \times k \times k}$ and input $X \in \mathbb{R}^{M \times H \times W}$, where $N$ and $M$ denote the number of output channels and input channels, $k$ is kernel size, and $H \times W$ is the size of the input image along each channel. Based on a common technique *im2col* (Chellapilla et al., 2006), the convolution layer can be converted to matrix multiplication as $\overline{Y} = \overline{W} \cdot \overline{X}$, where $\overline{W} \in \mathbb{R}^{N \times Mk^2}$ and $\overline{X} \in \mathbb{R}^{Mk^2 \times HW}$. For kernel decorrelation, we apply singular value decomposition (SVD) to map kernels into orthogonal space as: $\overline{W} = \sum_{i=1}^{N} \sigma_i \cdot \boldsymbol{u}_i \cdot \boldsymbol{v}_i^T$, where $\{\boldsymbol{u}_i\}_{i=1}^{N}$, $\{\boldsymbol{v}_i\}_{i=1}^{N}$ are two sets of orthogonal vectors[2]. The convolution can be decomposed as

$$\overline{Y} = \sum_{i=1}^{N} \overline{Y}_i = \sum_{i=1}^{N} \sigma_i \cdot \boldsymbol{u}_i \cdot \boldsymbol{v}_i^T \cdot \overline{X}. \tag{1}$$

For $\forall i \neq j$, it is easy to verify that $\langle \overline{Y}_i, \overline{Y}_j \rangle = \sigma_i \cdot \sigma_j \cdot tr(\overline{X}^T \cdot \boldsymbol{v}_i \cdot \boldsymbol{u}_i^T \cdot \boldsymbol{u}_j \cdot \boldsymbol{v}_j^T \cdot \overline{X}) = 0$, namely the output features $\overline{Y}_i$ and $\overline{Y}_j$ are orthogonal. Therefore, if we regard $\overline{W}_i = \sigma_i \cdot \boldsymbol{u}_i \cdot \boldsymbol{v}_i^T$ as a principal kernel, different principal kernels create orthogonal output features. To illustrate this, Figure 3 shows an input image (left) and the outputs (right three) generated by principal kernels. We can observe that principal kernels captures different features and serve different purposes.

As revealed in (Xie et al., 2017; Balestriero et al., 2018; Wang et al., 2020), imposing orthogonality on kernels leads to better training performance. This motivates us to initiate the training with a set of orthogonal kernels. Furthermore, to preserve kernel orthogonality during training, it is critical to constantly refresh the orthogonal space through re-decomposition. The above intuitions based on the observation on orthogonality play a key role in PriSM, which is described in the following section.

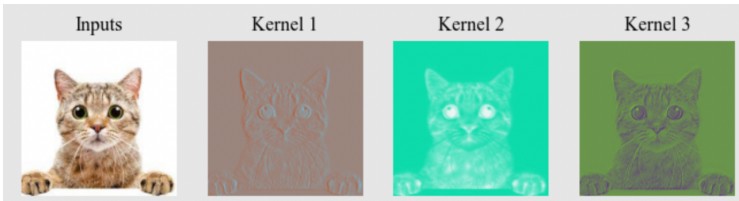

Figure 3: Orthogonal features generated by principal kernels. Different principal kernels capture different features (Kernel 1: outline of the cat, Kernel 2 and 3: textures but on distinct regions).

**Remark 3.1.** *The layer decomposition using SVD also applies to general linear layers (e.g., MLP, LSTM), where weights $W \in \mathbb{R}^{N \times M}$. We can directly apply SVD and obtain the principal components.*

### 3.2 PRISM: PRINCIPAL RANDOM SUB-MODEL TRAINING

Motivated by the observation that output features are orthogonal given orthogonal kernels, we propose PriSM, a new sub-model training method that directly trains orthogonal kernels in clients. Particularly, PriSM introduces two key components to ensure training performance with sub-models under very constrained settings. First, considering different orthogonal kernels and their contributions to outputs are weighed by the corresponding singular values, PriSM devises an importance-aware sampling scheme to create client sub-models to achieve computation and communication efficiency. In particular, the sampling scheme brings two benefits: i) each sub-model is a low-rank approximation of the server model; ii) the conglomerate of the sampled sub-models enables a nearly full converge of

---

[2]We assume w.l.o.g $\overline{W}$ is a tall matrix.

orthogonal kernels. Second, to further improve the memory efficiency while maintaining training performance, PriSM exploits low-rank structure in activations and allows each client to learn only a subset of intermediate representations in each layer. We start with unfolding sub-model creation, and then describe the training procedure in PriSM.

### 3.2.1 SUB-MODEL CREATION

**Computation and communication efficiency via sub-model sampling.** For a convolution layer with principal kernels $\left\{\overline{W}_i\right\}_{i=1}^N$, and the corresponding singular values $\{\sigma_i\}_{i=1}^N$, we randomly sample $r$ principal kernels denoted by $\overline{W}^c$ with sampling probability for $i$-th kernel as follows:

$$p_i = \frac{\sigma_i^\kappa}{\sum_{j=1}^N \sigma_j^\kappa}. \tag{2}$$

Here, $r$ in each layer is decided by $c$-th client's resource budget, $\kappa$ is a smooth factor controlling the probability distribution in sampling. Indices of selected kernels are denoted as $\mathcal{I}_c$.

The convolution output is calculated as

$$\overline{Y} = \sum_{i \in \mathcal{I}_c} \overline{Y}_i = \sum_{i \in \mathcal{I}_c} \sigma_i \cdot \boldsymbol{u}_i \cdot \boldsymbol{v}_i^T \cdot \overline{X}. \tag{3}$$

By performing the sampling process for every convolution layer, we create a *random* low-rank model for each client. *Important* kernels with large singular values are more likely to be chosen, and all sub-models can together provide nearly full coverage of principal kernels. The resulting sub-model hence has convolution layers with fewer kernels, reducing required computations and communication overhead (when aggregating models). Other element-wise layers, such as ReLU and batch normalization, remain the same.

**Memory efficiency via learning a subset of features.** We further exploit low-rank structure in intermediate representations. For a sub-model above, each convolution layer is decomposed into two sublayers: $Conv_U$ and $Conv_V$ with kernels $\{\boldsymbol{u}_i\}_{i \in \mathcal{I}_c}$ and $\{\boldsymbol{v}_i\}_{i \in \mathcal{I}_c}$ respectively. Without further optimization, output from $Conv_U$ consumes the same memory as the original layer, causing high memory pressure at the edge. However, noting that inputs to $Conv_U$, $\overline{X} \in \mathbb{R}^{r \times HW}$, is low-rank, and so kernels $\overline{W} \in \mathbb{R}^{N \times r}$, based on the rank inequality of matrix multiplication (Banerjee & Roy, 2014), we obtain

$$\text{Rank}(\overline{Y}) \leq \min\left(\text{Rank}(\overline{X}), \text{Rank}(\overline{W})\right) \leq r. \tag{4}$$

Therefore, output of $Conv_U$ also exhibits a low-rank structure. Such a low-rank structure reflects correlation among output channels, indicating channel redundancy in outputs when $r < N$. Based on the observation, we further allow $Conv_U$ to compute only a subset of output channels, therefore reducing memory footprints of intermediate representations while still preserving necessary information in activations. Typically, given input to $Conv_U$ with $r$ input channels, PriSM only computes $r$ output channels. During implementation, we replace the select $\boldsymbol{u}_i$ with its subset $\boldsymbol{u}_i(1:r)$ when computing output features from $Conv_U$. For simplicity, we will still use $\boldsymbol{u}_i$ in the rest of the paper to denote the subset $\boldsymbol{u}_i(1:r)$. The unselected channels are cached in $\hat{\boldsymbol{u}}_i$, and are used in model aggregation.

### 3.2.2 TRAINING

This section details the training procedure in PriSM. We describe each component below.

**Local training.** On each client, during local training, the sub-model with parameter $\{\sigma_i, \boldsymbol{u}_i, \boldsymbol{v}_i\}_{i \in \mathcal{I}_c}$ are updated. The sub-model also consists of trainable layers such as BatchNorm but with fewer channels denoted in $\mathcal{I}_c$. PriSM allows $\sigma_i$ to be trained in local training, thus ensuring changes regarding each principal kernel's importance are captured in local clients. In addition, $\sigma_i$ will be merged into $\boldsymbol{u}_i, \boldsymbol{v}_i$ as $\boldsymbol{u}_i' = \sqrt{\sigma_i}\boldsymbol{u}_i, \boldsymbol{v}_i' = \sqrt{\sigma_i}\boldsymbol{v}_i$ to reduce memory footprints.

**Sub-model aggregation.** On the server side, with sub-models obtained from clients, we aggregate $i$-th principal kernel as follows:

$$\overline{W}_i = \left(\left[\sum_{c \in \mathcal{C}} \alpha_i^c \boldsymbol{u}_i'^c; \quad \hat{\boldsymbol{u}}_i\right]\right) \cdot \left(\sum_{c \in \mathcal{C}} \alpha_i^c \boldsymbol{v}_i'^c\right)^T, \tag{5}$$

where $\mathcal{C}$ denotes the subset of active clients, $\alpha_i$ is the aggregation coefficient for $i$-th kernel. We propose a weighted averaging scheme: if $i$-th kernel is selected and trained by $C_i$ clients, then $\alpha_i^c = 1/C_i$. Furthermore, the unselected output channels $\hat{u}_i$ are concatenated with the aggregated $u_i$ before reconstructing the orthogonal kernel to preserve model capacity. If $\overline{W}_i$ was not selected by any client, it remained unchanged in the server. The full model in the original space is constructed by converting each 2-dimensional $\overline{W}_i$ to the original dimension $\mathbb{R}^{M \times k \times k}$ and combining them.

**Orthogonal space refresh.** After model aggregation, we perform SVD on the updated kernels $\overline{W}$ to preserve orthogonality among the principal kernels. Thus, in the next communication round, the importance-aware sampling can still create low-rank sub-models for different clients..

We further use two additional techniques to improve learning efficiency in the orthogonal space: activation normalization, and regularization on orthogonal kernels.

**Activation normalization.** We apply batch normalization without tracking running statistics; namely, the normalization always uses current batch statistics in the training and evaluation phases. Each client applies normalization separately with no sharing of statistics during model aggregation. Such an adaptation is effective in ensuring consistent outputs between different sub-models and avoids potential privacy leakage through the running statistics (Andreux et al., 2020).

**Regularization.** When learning a factorized model on a client, applying weight decay to $u_i$ and $v_i$ separately results in poor final accuracy. Inspired by (Khodak et al., 2020), for training on client $c$, we add regularization to the subset of kernels as follows:

$$reg = \frac{\lambda}{2} \left\| \sum_{i \in \mathcal{I}_c} u_i' \cdot v_i'^{T} \right\|_F^2 , \tag{6}$$

where $\lambda$ is the regularization factor, $\mathcal{I}_c$ denotes the subset of principal kernels on client $c$.

Algorithm 1 presents an overall description of PriSM. We only show the procedure on a single convolution layer with kernels $W$ for the sake of simplifying notations.

---

**Algorithm 1** PriSM: Principal Random Sub-Model Training

---

**Input:** layer parameters $W$, client capacities.

 1: **for** communication round $t = 1, \cdots, T$ **do**
 2:     Decompose $W$ into orthogonal kernel using SVD $\rightarrow \left\{ \overline{W}_i \right\}_{i=1}^{N}$.
 3:     Choose a subset of clients $\rightarrow \mathcal{C}$.
 4:     **for** each client $c \in \mathcal{C}$ **do**
 5:         Compute the sub-model size for client $c \rightarrow |\mathcal{I}_c|$.
 6:         Obtain a sub-model following the procedure in Sec 3.2.1 $\rightarrow \mathcal{I}_c, \overline{W}^c$. // Sub-model
 7:         Perform **LocalTrain** $\leftrightarrow \mathcal{I}_c, \overline{W}^c$. // Local training
 8:     **end for**
 9:     Aggregate parameters based on Eq. (5) $\overline{W} \leftarrow \left\{ \overline{W}^c \right\}_{c \in \mathcal{C}}$. // Sub-model aggregation
10:     Perform SVD on $\overline{W}$. // Orthogonal space refresh
11: **end for**

---

12: **LocalTrain** $\leftrightarrow \mathcal{I}_c, \overline{W}^c$
13: **for** local iteration $k = 1, \cdots, K$ **do**
14:     Sample an input batch from the local dataset $\rightarrow \mathcal{D}_k$.
15:     Perform the forward and backward pass $\leftarrow \mathcal{D}_k, \overline{W}^c$.
16:     Update the local sub-model using SGD $\rightarrow \overline{W}^c$.
17: **end for**

---

In the following remarks, we differentiate PriSM from Dropout and Low-Rank compression.

**Remark 3.2. *PriSM vs Dropout.*** *PriSM shares some computation similarity with model training using regular dropout in clients. However, regular dropout suffers convergence instability due to*

*inconsistent activations across iterations, esp. with a high dropout probability (Horvath et al., 2021). In contrast, PriSM performs importance-aware sampling in the orthogonal space. Each sub-model approximates to the full model, and different sub-models do not create significant inconsistency.*

**Remark 3.3.** ***PriSM vs Low-Rank Compression.*** *PriSM is not a low-rank compression method. Low-rank compression methods such as FedHM (Yao et al., 2021) aim to construct a smaller server model by completely discarding some kernels even though they can still contribute to training performance (See Figure 6 in Section 4.3). PriSM randomly select sub-models so that every kernel is possible to be learned. Furthermore, PriSM achieves memory efficiency by exploiting low-rank properties in intermediate representations, which is not seen in other low-rank methods.*

## 4 EXPERIMENTS

We evaluate PriSM under resourced-constrained settings where no clients can training the large full model. Furthermore, we consider both homogeneous and heterogeneous client settings. Specifically, in homogeneous settings, all clients have the same limited computation, memory, and communication capacity, while in heterogeneous settings, clients' capacities might vary. We also compare PriSM with two other baselines: ordered dropout in orthogonal space (OrthDrop); and ordered dropout in original space (OrigDrop). At a high level, our results demonstrate that PriSM achieves comparable server model accuracy even when only training very small sub-models on all clients. Additionally, we provide more insights into the superior performance of PriSM by analyzing server model's rank structure. Finally, we study the sampling process and cost breakdown in the FL system.

**Baselines.** Prior methods such as FjORD (Horvath et al., 2021) and HeteroFL (Diao et al., 2021) select sub-models from the original kernel space, for which we denote as OrigDrop. In implementation, we follow the same procedure in HeteroFL. On the other hand, we use OrthDrop to denote selecting fixed top-k kernels from the orthogonal space such as in FedHM (Yao et al., 2021).

**Models and Datasets.** We train ResNet-18 on CIFAR-10 (Krizhevsky et al., 2009), CNN on FEM-NIST (Caldas et al., 2018) and LSTM model on IMDB (Maas et al., 2011)[3]. Detailed architectures are provided in Appendix A.1.1.

**Data Distribution.** For CIFAR-10 and IMDB, we uniformly sample an equal number of training images for each client when creating i.i.d datasets. For non-i.i.d datasets, we first use Dirichlet function $Dir(\alpha)$ (Reddi et al., 2020) to create sampling probability for each client and then sample an equal number of training images for clients. We create two different non-i.i.d datasets with $\alpha = 1$ and $\alpha = 0.1$, where a smaller $\alpha$ indicates a higher degree of non-i.i.d. For FEMNIST, we directly use the dataset without any additional preprocessing.

**FL Setting.** We simulate an FL setting with 100 clients with 20 random clients active in each communication round. Each client trains its model for 2 local epochs in each round. We use SGD with momentum during training. The learning rate is initially $0.1$ for ResNet-18 and $0.01$ for CNN/FEMNIST, and decayed by a cosine annealing schedule. Further details are in Appendix A.1.2.

### 4.1 PERFORMANCE ON CONSTRAINED HOMOGENEOUS CLIENTS

In this setting, we assume that all clients have the same limited capacity. We vary the client sub-model size from $0.2$ to $0.8$ of the full server model, where $0.x$ indicates only a $0.x$ subset of the principal kernels are sampled in each convolution layer from the server model (denoted as *keep ratio*). Accordingly, $r = 0.x \cdot N$, where $N$ is the number of output channels in a layer. In Table 1, we list the computation, memory and communication footprints for different sub-models of ResNet-18. CNN/FEMNIST and LSTM/IMDB also enjoy similar cost reductions. It is worth noting that PriSM incurs much smaller costs compared to FedHM (See Appendix A.4 for further details).

Figure 4 shows final validation accuracy drops of ResNet-18 with different sub-models on i.i.d and non-i.i.d datasets compare to full-model training. We note that PriSM constantly delivers better performance than the other two baselines. The performance gap is even more striking under very constrained settings. For instance, when only $0.2\times$ sub-models are supported on clients, PriSM attains comparable accuracy as full-model training, and achieves up to $10\%/14\%$ performance improvement

---

[3]Due to the page limit, LSTM results are deferred to Appendix A.2

Table 1: Model size, MACs, activation memory for different sub-models in PriSM (batch size: 32).

| Model | | Full | 0.8 | 0.6 | 0.4 | 0.2 |
|---|---|---|---|---|---|---|
| | Params | 11 M | 7.9 M (72%) | 4.5 M (41%) | 2 M (18%) | .5 M (4.5%) |
| ResNet-18 | MACs | 35 G | 25.5 G (73%) | 14.7 G (42%) | 6.87 G (20%) | 2 G (5.6%) |
| | Mem | 31.5 M | 37 M (115%) | 28.3 M (90%) | 18.9 M (60%) | 9.5 M (30%) |

compared to OrthDrop and OrigDrop on non-i.i.d dataset with $\alpha = 0.1$. Furthermore, for $0.2\times$ sub-models, if slightly relaxing memory constraints and allowing PriSM to learn $2\times$ out channels in $Conv_U$ (denoted as PriSM-O2), the accuracy can be further improved with only incurring additional 65% memory footprint. We also make two key observations. First, training with sub-models in the orthogonal space provides better performance than in the original space, which aligns with our intuition in Section 3.1. Second, our importance-aware sampling strategy for creating sub-models is indispensable as demonstrated by the notable performance gap between PriSM and OrthDrop. Similar performance trends are also observed in FEMNIST, as shown in Figure 5.

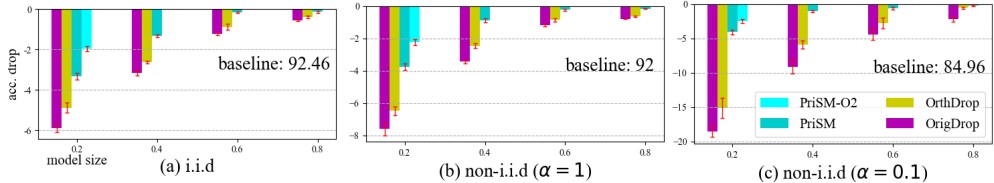

Figure 4: Accuracy drops on CIFAR-10 on homogeneous clients. PriSM constantly delivers better performance compared to OrigDrop and OrthDrop, significantly outperforming them under very constrained realistic edge settings. (bar: mean; line: std)

## 4.2 PERFORMANCE ON CONSTRAINED HETEROGENEOUS CLIENTS

To simulate clients with varying limited capacity, we simulate the following setting: 40% clients train $0.4\times$ sub-models, and 60% clients train $0.2\times$ sub-models. No participating client trains the full model. For baseline methods, we follow the same strategy as in Section 4.1.

Table 2 lists the final accuracy achieved by different methods under the heterogeneous setting. PriSM greatly outperforms the baseline methods even when $0.4\times$ sub-models are supported on a small fraction of clients. Furthermore, similar to the results in Section 4.1, the benefits of training in the orthogonal space and importance-aware sampling strategy are also observed in heterogeneous client settings.

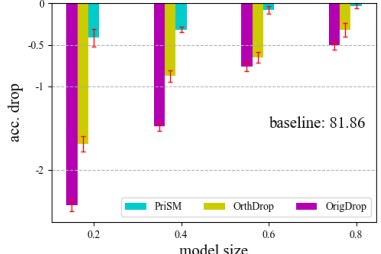

Figure 5: Accuracy drops on FEMNIST on homogeneous clients. (bar: mean; line: std)

Table 2: Training performance of ResNet-18/CIFAR-10 on heterogeneous clients.

| Distribution | Baseline | OrigDrop | OrthDrop | PriSM |
|---|---|---|---|---|
| i.i.d | 92.46 | $88.86 \pm 0.17$ | $89.57 \pm 0.16$ | $90.63 \pm 0.14$ |
| non-i.i.d (1) | 92 | $86.91 \pm 0.24$ | $88.78 \pm 0.22$ | $89.89 \pm 0.2$ |
| non-i.i.d (0.1) | 84.96 | $76.38 \pm 0.92$ | $78.37 \pm 0.99$ | $82.58 \pm 0.51$ |

## 4.3 INSIGHTS INTO PRISM

We now focus on providing further insights into PriSM by analyzing some of its key aspects. To this end, we first examine the low-rank structure of models during training, and pinpoint the cause behind the accuracy gap between fixed and random kernel dropout strategies in the orthogonal space. Thereafter, we study the sampling process and cost breakdown in the FL system

**Model's rank during training.** To analyze the server model's rank structure, we adopt a similar method as in Alter et al. (2000) to calculate the required number of principal kernels to accurately

approximate each layer as $2^{-\log\left(\sum_i p_i \log p_i\right)}$. Here, $p_i$ is calculated as in Eq. (2) with $\kappa = 1$. Figure 6 shows the number of required kernels for each layer in ResNet-18 during full-model FL (Block$i$-$j$: $j$-th convolution layer in $i$-th ResBlock). We first observe that while the server model attains a low-rank structure gradually, a randomly initialized model does not. Therefore, sub-models with fixed top-k principal kernels inevitably causes reductions in the server model capacity. Furthermore, even at the end of the FL training, around half principal kernels are still required to approximate most layers. In fact, some layers require even more principal kernels. Therefore, our probabilistic sampling scheme is essential in preserving the server model capacity during FL training with sub-models.

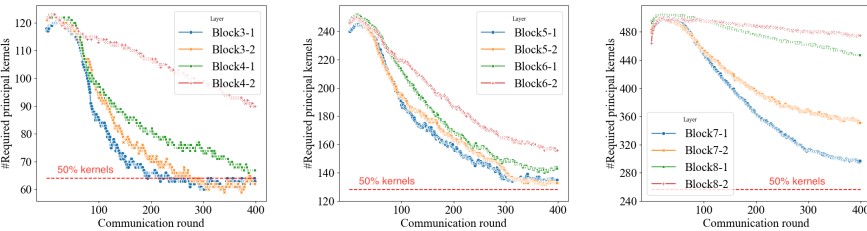

(a) ResBlock 3, 4 (128 kernels). (b) ResBlock 5, 6 (256 kernels). (c) ResBlock 7, 8 (512 kernels).

Figure 6: The number of principal kernels required to accurately approximate each convolution layer in ResBlocks 3-8 in ResNet-18 (Results of ResBlocks 1 and 2 are discussed in Figure 1).

**Kernel sampling profiling.** Figure 7 shows the average number of clients assigned for each orthogonal channel in one communication round. Each client trains a $0.2\times$ sub-model of ResNet-18 on CIFAR-10 with i.i.d distributions as in Sec 4.1. We observe that each kernel is selected by at least one client in each round, indicating every kernel will be activated and trained on local data in each round. Furthermore, orthogonal kernels with larger $\sigma$ get more chances to be chosen, which ensures sub-models on all clients consistently approximate the full model.

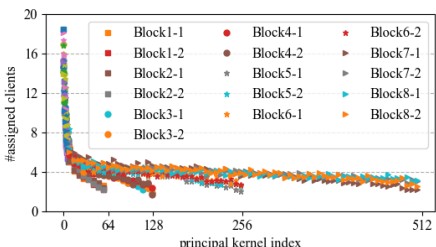

Figure 7: The average number of clients assigned for each orthogonal kernel during training.

**Runtime breakdown.** In this section, we investigate the relative cost of SVD by breaking down the model training time into four stages: sub-model creation, local training, model aggregation, and obtaining orthogonal kernels (SVD). We choose a large model, ResNet-18, as the target model. We adopt the same hyperparameters as in Table 7. We run the server and clients process on NVIDIA Quadro RTX 5000. Table 3 lists each stage's average time in one communication round. We observe that the time of SVD is nearly negligible compared to the time spent on the local training. Therefore, even for large models such as ResNet-18, the relative overhead of SVD is still very small.

Table 3: Training time breakdown for ResNet-18 on CIFAR-10.

| stage | sub-model creation | local training | aggregation | SVD |
|-------|--------------------|----------------|-------------|-----|
| time  | 3.27 s             | 36.82 s        | 0.11 s      | 0.96 s |

## 5  CONCLUSION

We have considered the practical, yet under-explored, problem of federated learning in a resource-constrained edge setting, where no participating client has the capacity to train a large model. As our main contribution, we propose the PriSM training methodology, that empowers the resource-limited clients by enabling them to train smaller sub-models. At the same time, PriSM utilizes a novel sampling approach to obtain sub-models for the clients, all of which together ensure that the server model achieves close to the full-model performance. PriSM further improves memory efficiency by exploiting low-rank structure in intermediate activations. Our extensive empirical results demonstrate that PriSM performs significantly better than the prior baselines, especially when each client can train only a very small sub-model. We will consider providing theoretical analysis in the future work.

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

## A APPENDIX

### A.1 MODELS AND HYPERPARAMETERS

In this section, we provide detailed information about models and hyperparameter settings for the results presented in the paper. We will open our source code upon acceptance of the paper.

#### A.1.1 MODELS

**ResNet-18/CIFAR-10.** We use a ResNet-18 optimized for CIFAR-10, in which kernel size in the first convolution layer is changed from $7 \times 7$ to $3 \times 3$. Details are shown in Table 4.

Table 4: ResNet-18/CIFAR-10

| Module | | #kernels | size | stride | Batch Norm | ReLU | Downsample |
|---|---|---|---|---|---|---|---|
| Conv1 | | 64 | 3 | 1 | ✓ | ✓ | ✗ |
| ResBlock 1 | Block1-1 | 64 | 3 | 1 | ✓ | ✓ | ✗ |
| | Block1-2 | 64 | 3 | 1 | ✓ | ✓ | |
| ResBlock 2 | Block2-1 | 64 | 3 | 1 | ✓ | ✓ | ✗ |
| | Block2-2 | 64 | 3 | 1 | ✓ | ✓ | |
| ResBlock 3 | Block3-1 | 128 | 3 | 1 | ✓ | ✓ | ✓ |
| | Block3-2 | 128 | 3 | 1 | ✓ | ✓ | |
| ResBlock 4 | Block4-1 | 128 | 3 | 1 | ✓ | ✓ | ✗ |
| | Block4-2 | 128 | 3 | 1 | ✓ | ✓ | |
| ResBlock 5 | Block5-1 | 256 | 3 | 1 | ✓ | ✓ | ✓ |
| | Block5-2 | 256 | 3 | 1 | ✓ | ✓ | |
| ResBlock 6 | Block6-1 | 256 | 3 | 1 | ✓ | ✓ | ✗ |
| | Block6-2 | 256 | 3 | 1 | ✓ | ✓ | |
| ResBlock 7 | Block7-1 | 512 | 3 | 1 | ✓ | ✓ | ✓ |
| | Block7-2 | 512 | 3 | 1 | ✓ | ✓ | |
| ResBlock 8 | Block8-1 | 512 | 3 | 1 | ✓ | ✓ | ✗ |
| | Block8-2 | 512 | 3 | 1 | ✓ | ✓ | |
| Classification | | 10 | - | - | ✗ | ✗ | ✗ |

**CNN/FEMNIST.** We use a similar architecture as in FjORD Horvath et al. (2021). The detailed model is shown in Table 5.

Table 5: CNN/FEMNIST

| Module | #kernels | size | stride | ReLU |
|---|---|---|---|---|
| Conv1 | 64 | 5 | 1 | ✓ |
| Pooling1 | - | 2 | 2 | ✗ |
| Conv12 | 64 | 3 | 1 | ✓ |
| Pooling2 | - | 2 | 2 | ✗ |
| Classification | 10 | - | - | ✗ |

**LSTM/IMDB.** We use a common LSTM model as shown in Table 6.

#### A.1.2 TRAINING HYPERPARAMETERS

**ResNet-18/CIFAR-10 on homogeneous clients.** We simulate 100 clients during FL training, in which each client is assigned 500 training samples for both i.i.d and non-i.i.d datasets. In each

Table 6: LSTM/IMDB

| Module | input size | output size | hidden size | #layers |
|--------|-----------|-------------|-------------|---------|
| Embedding | 1001 | 64 | - | - |
| LSTM | 64 | 256 | 256 | 2 |
| FC | 256 | 1 | - | - |

communication round, each client performs local training for 2 epochs using the local data, then uploads parameters to the server for aggregation. Table 7 lists detailed hyperparameters during FL training with ResNet-18.

Table 7: Hyperparameters for ResNet-18/CIFAR-10 on homogeneous clients

| Datasets | #clients | #samples | distribution | | augmentation |
|----------|----------|----------|--------------|--|--------------|
| | 100 | 500 | i.i.d, non-i.i.d ($\alpha = 1, 0.1$) | | flip, random crop |
| Training | #Rounds | #local epochs | batch size | #active clients | smooth factor $\kappa$ |
| | 1000 | 2 | 32 | 20 | 2.5/4 (0.2 sub-model) |
| Opt | Optimizer | Momentum | $wd$ | initial $lr$ | scheduler |
| | SGD | 0.9 | 0.0002 | 0.1 | cosine annealing |

**CNN/FEMNIST on homogeneous clients.** We simulate 100 clients during FL training, in which each client is assigned 10 users' data from the original training dataset. We use the whole validation dataset to compute the validation accuracy. Table 8 lists detailed hyperparameters during FL training with CNN.

Table 8: Hyperparameters for CNN/FEMNIST on homogeneous clients

| Datasets | #clients | #users/client | distribution | | augmentation |
|----------|----------|---------------|--------------|--|--------------|
| | 100 | 10 | natural non-i.i.d | | None |
| Training | #Rounds | #local epochs | batch size | #active clients | smooth factor $\kappa$ |
| | 100 | 2 | 32 | 20 | 2.5 |
| Opt | Optimizer | Momentum | $wd$ | initial $lr$ | scheduler |
| | SGD | 0.9 | 0.0002 | 0.01 | cosine annealing |

**ResNet-18/CIFAR-10 on heterogeneous clients.** We adopt the same setting as in Table 7, except the fact that clients might vary in computation and communication capacity. Therefore different model might train sub-models with different sizes (See Sec 4.2 in the main paper).

## A.2 EXPERIMENTS ON LSTM/IMDB

The LSTM model used in FL training is detailed in Table 6. During training, we simulate 100 clients, in which each client is assigned 375 training samples. We create local datasets with two different distributions using the same method as in CIFAR-10: i.i.d and non-i.i.d ($\alpha = 0.1$). Table 9 list detailed hyperparameters for training LSTM/IMDB.

Figure 8 shows accuracy drops of sub-model training on IMDB in homogeneous client settings. While the task on IMDB is just a binary classification problem, PriSM still achieves the best final server model accuracy on both i.i.d and non-i.i.d datasets. On i.i.d datasets, training using sub-models achieves comparable accuracy as full-model training, even only using very small sub-models such as $0.2\times$. On the other hand, on non-i.i.d dataset, OrigDrop suffers notable accuracy drops compared to PriSM.

Table 9: Hyperparameters for LSTM/IMDB on homogeneous clients

| Datasets | #clients | #samples | distribution | | augmentation |
|---|---|---|---|---|---|
| | 100 | 375 | i.i.d, non-i.i.d ($\alpha = 0.1$) | | None |
| Training | #Rounds | #local epochs | batch size | #active clients | smooth factor $\kappa$ |
| | 300 | 2 | 32 | 20 | 2 |
| Opt | Optimizer | Momentum | $wd$ | initial $lr$ | scheduler |
| | SGD | 0.9 | 0.0002 | 0.1 | cosine annealing |

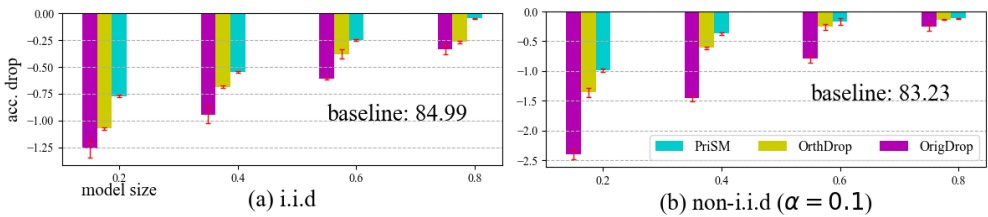

Figure 8: Accuracy drops on IMDB on homogeneous clients.

## A.3 MORE ABLATION STUDY

In this section, we study the effects of two hyperparameters: the number of local epochs and active clients. Each of these two parameters affects the sampling process when creating sub-models for clients. Specifically, given a fixed number of total iterations, FL training with a small number of local epochs per communication round performs a more frequent sampling process, thus making more orthogonal kernels to be selected and trained. Similarly, the FL training with a large number of active clients per round can also activate more orthogonal kernels.

**Effects of number of local epochs.** To investigate the effects of the number of local epochs on the final server model accuracy, we train ResNet-18 in homogeneous client settings. The sub-model trained on clients varies from $0.2\times$ to $0.6\times$. We also train a full model as a baseline. The common training hyperparameters are the same as in Table 7. We fix the number of total iterations as 2000, namely, Round $\times$ Local epochs $= 2000$. Figure 9 shows final server model accuracy on i.i.d and non-i.i.d datasets with $\alpha = 0.1$. We observe that while final server model accuracy decreases as the number of local epochs increases, the accuracy gap between sub-model and full-model training also slightly increases. One potential reason is that the total number of sampling decreases in FL training with more local epochs. As a result, some orthogonal kernels are under-trained. Such observation also aligns with the intuition discussed in the main paper that all orthogonal kernels should be trained.

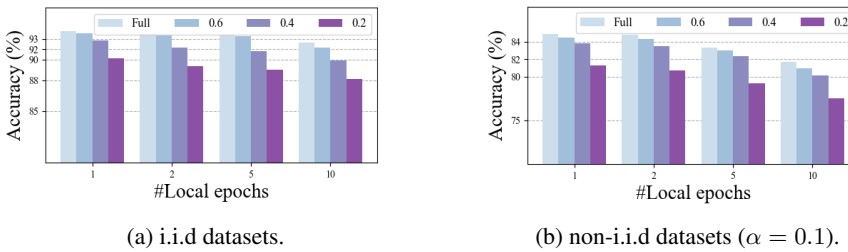

(a) i.i.d datasets.                (b) non-i.i.d datasets ($\alpha = 0.1$).

Figure 9: Effects of the number of local epochs on training performance (ResNet-18/CIFAR-10).

**Effects of number of active clients.** We follow the same settings as in Table 7 except that the number of active clients varies from 10 to 30 in each communication round. Similar as above, we vary the sub-model trained on clients from $0.2\times$ to $0.6\times$. Figure 10 shows the final server model accuracy on i.i.d and non-i.i.d datasets with $\alpha = 0.1$. On i.i.d datasets, FL training with a different number of active clients does not significantly change the training performance. Furthermore, the accuracy gap between sub-model and full-model training is also almost the same, even though a less frequent sampling process is performed in training with fewer clients. However, on non-i.i.d datasets, the finaly accuracy increases with an increasing number of participating clients. Further, the performance gap between sub-model and full-model training also shrinks as more sampling processes are performed in each round. In practical edge settings, with plenty of devices such as smart-home devices connected, the performance gap can be possibly further shrunk.

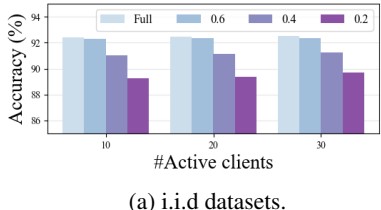

(a) i.i.d datasets.

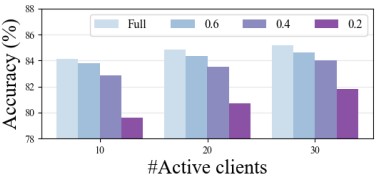

(b) non-i.i.d datasets ($\alpha = 0.1$).

Figure 10: Effects of the number of active clients on training performance (ResNet-18/CIFAR-10).

## A.4 COST COMPARISON BETWEEN PRISM AND FEDHM

In this section, we present a comparison of PriSM and FedHM on sub-model size (#Params), computations (#MACs), and memory for intermediate features (Mem). For computations and memory, we only consider the forward pass during training as it is sufficient to show the difference between these two methods. When measuring memory requirements, we ignore data representations (float/fixed point) and only count the number of values in activations. Table 10 lists the comparison. It is noted that FedHM still needs significant memory to store activations, even for very small sub-models. PriSM reduces both model size, computations, and activations when using small sub-models.

Table 10: Costs for different sub-models in PriSM and FedHM (batch size: 32).

| Model | | | Full | 0.8 | 0.6 | 0.4 | 0.2 |
|---|---|---|---|---|---|---|---|
| | | | | PriSM | | | |
| | Params | | 11 M | 7.9 M (72%) | 4.5 M (41%) | 2 M (18%) | .5 M (4.5%) |
| ResNet-18 | MACs | | 35 G | 25.5 G (73%) | 14.7 G (42%) | 6.87 G (20%) | 2 G (5.6%) |
| | Mem | | 31.5 M | 37 M (115%) | 28.3 M (90%) | 18.9 M (60%) | 9.5 M (30%) |
| | | | | FedHM | | | |
| | Params | | 11 M | 9.9 M (90%) | 7.4 M (67%) | 4.9 M (44%) | 2.5 M (22%) |
| ResNet-18 | MACs | | 35 G | 28.8 G (82%) | 22.4 G (64%) | 16 G (46%) | 7.4 G (23%) |
| | Mem | | 31.5 M | 44.9 M (143%) | 40.5 M (129%) | 36.1 M (115%) | 31.7 M (101%) |

## A.5 MORE ON-DEVICE EFFICIENCY ANALYSIS

Table 11: Per-round of training with different sub-models.

| sub-model | full | $0.8\times$ | $0.6\times$ | $0.4\times$ | $0.2\times$ |
|---|---|---|---|---|---|
| time | 31.41 s | 28.39 s | 23.53 s | 20.21 s | 17.55 s |

We further looked into how computation and memory savings result in clock-time improvements. We use ResNet-18 and train it on CIFAR-10. In each communication round, each client trains its

sub-model with 1 epoch with batch size 32. Other hyperparameters are the same as in Table 7. We use a single NVIDIA GTX 1080 GPU as the test platform. Table 11 shows per-round time on local training with different sub-models. We observe that per-round clock time decreases with the size of sub-models on clients. Therefore, this computation and memory saving indeed leads to clock-time improvements.

