# OpenReview forum: "Federated Learning of Large Models at the Edge via Principal Sub-Model Training"
_ICLR.cc/2023/Conference — Submitted to ICLR 2023_

### Official Review · Reviewer_v26F · 2022-10-25

**Confidence:** 3
**Correctness:** 4
**Technical Novelty And Significance:** 3
**Empirical Novelty And Significance:** 3
**Recommendation:** 6

**Clarity, Quality, Novelty And Reproducibility:**

This paper is well-organized and the PriSM approach seems to be reproducible with a clear instruction on the hyper-parameters.

**Strength And Weaknesses:**

Strength
1. This paper is well-written with clear formulation and introduction of PriSM

2. The empirical evaluation of ResNet models is solid across different vision FL datasets.

3. In the appendix, the paper presents a detailed parameter selection, making PriSM reproducible

Weakness
1. A stronger analysis of the on-device efficiency should be introduced.

2. It would be better for PriSM to be applied to general linear models instead of focusing on convolution blocks.

**Summary Of The Paper:**

This paper studies the problem of federated learning on edge devices with limited computation, memory, and communication capacities. Specifically, this paper proposed PriSM, a sub-model training approach for FL. PriSM is a low-rank approach that performs SVD on the kernels to build a sub-model on device during the training.  With empirical evaluation, the authors show that PriSM is able to train a model with only 20% of the principle kernels and has 10% performance improvements over existing solutions.


**Summary Of The Review:**


This paper proposes a powerful low-rank approach in FL for sub-model training on-device. I have already mentioned the strengths in the above section. Here I would like to raise a few questions for clarification.

1. What is the on-device computation efficiency? How many FLOPs does the PriSM save when performing FL? Does this saving result in clock-time efficiency improvements?

2.  Does PriSM generalize to MLP/Transformer models? The linear model can also be low-rank decomposed on-device for fast training.

3. Is there any extra overhead in the aggregation phase? How does this overhead compare with the original aggregation procedure in FedAvg?

---

### Official Review · Reviewer_vmY3 · 2022-10-25

**Confidence:** 3
**Correctness:** 3
**Technical Novelty And Significance:** 2
**Empirical Novelty And Significance:** 3
**Recommendation:** 6

**Clarity, Quality, Novelty And Reproducibility:**

I believe the paper can be improved in terms of writing and presentation. I had difficulty to understand some parts of the paper due to quality of writing. Specifically, I think authors can provide more detailed discussions on local training part on page 5. I am not clear that what it means that ``the selected $\sigma_i,u_i, v_i  i\in\mathcal I_c$ are updated, together with trainable parameters in other layers.'' If all trainable parameters in other layers need to be updated by the client how the proposed method help the clients with limited memory and computational capability?

**Strength And Weaknesses:**

Strengths:

1. The paper proposes a new method to enable clients with limited computational capability to train a large model.
2. Using the proposed method, clients with limited memory can collaborate on training a large model that cannot be fit into their memory.

Weaknesses:

1. This works lacks theoretical analysis. For example, it would be interesting to see if there are any other sampling method that can provide better result. Also it is not clear that how $r$ affects the training accuracy.
2. An important question that this paper does not respond is that why training a low rank representation of a large model is better than training of a original smaller model.



**Summary Of The Paper:**

The paper studies the problem of learning a large model in a federated manner by clients with limited computational and memory capabilities. The paper is mainly about large neural networks. The proposed solution is based on SVD decomposition of each layer to provide clients with a low rank representation of each layer. This reduces the amount of computations required to update the models locally by clients.

**Summary Of The Review:**

In summary, the paper studies an important problem and proposes a novel algorithm to solve this problem. However, there are important questions that should be answered and discussed by the paper. Also, writing and presentation of the paper need improvement.

---

> ### Comment · Reviewer_vmY3 · 2022-11-18
> **Discussion**
>
> Thanks authors for responding to my comments and revising the manuscript. I believe the revisions improved the paper and part of my concerns are addressed. I am happy to increase my score. However, I still believe that the paper lacks theoretical analysis of the proposed algorithm.

---

> > ### Author Response · Authors · 2022-11-19
> > **Thanks**
> >
> > We very much appreciate your support on the score. While the scope of the current manuscript is a proposal and empirical demonstration of a novel method for practical resource-constrained deployment of FL at scale, we take theoretical exploration as an important future research direction. Please let us know if you have further questions.

---

### Official Review · Reviewer_59uv · 2022-10-28

**Confidence:** 4
**Correctness:** 4
**Technical Novelty And Significance:** 2
**Empirical Novelty And Significance:** 2
**Recommendation:** 3

**Clarity, Quality, Novelty And Reproducibility:**

- The paper is easy to read and is of good quality.
- The novelty of the paper is limited since the main contribution appears to be a small tweak to an existing set of approaches in the literature.
- There are no links to any online code in the paper, which does not allow this reviewer to judge the reproducibility of this work.

**Strength And Weaknesses:**

**Strengths**

- The probabilistic selection of the principal kernels for the submodels using the strength of the singular values is an important insight that helps the authors with an improved algorithm.
- The authors have carried out extensive numerical experiments to highlight the advantages of their approach.

**Weaknesses**

- The paper has an incremental nature, with the major difference from existing literature being a focus on probabilistic sampling of the principal kernels. Perhaps the authors can look into some theoretical aspects of PriSM to overcome the limited innovation in terms of the algorithm. (_Note added after reading authors' response:_ The literature being referred to here is the general literature on compression of neural networks. Low-rank matrix factorization and other structured factorizations, as well as many other ad-hoc tricks, are routinely utilized in works that study compression of neural networks.)
- ~~The algorithm is very much limited to the case of convolutional neural networks and it is not clear how to adapt these ideas to the case of other architectures.~~ (_The reviewer agrees that some of the ideas in the paper are applicable to some other architectures._)

**Summary Of The Paper:**

The focus of this paper is on federated learning in a setting where the edge clients do not have sufficient resources to train a large model and, additionally, the clients do not want to share any intermediate data and/or labels with the server. The main contribution in the paper is an algorithm, termed Principal Sub-Model (PriSM), in which the convolution kernels are mapped to the so-called *principal kernels* given by the singular value decomposition (SVD) at the server and then clients are asked to update low-rank approximations of the convolution kernels that are given by a subset of the principal kernels. While such an idea exists in the literature, as noted in the paper, the main idea the authors leverage in this paper is to probabilistically decide on the principal kernels that correspond to the submodels at each client.

**Summary Of The Review:**

While the paper puts forth an effective idea, the contribution of the paper is limited and incremental. Such works are indeed valuable, but are perhaps better suited as workshop papers then as full conference papers.

---

**Response to the Authors**

Dear Authors,

I greatly regret I was not able to engage with you during Phase I of the discussion period due to emergency of a personal nature. Please accept my sincerest apologies for this, as you put in significant efforts into the work and you deserved a timely response.

I have gone through your response and have expanded / clarified parts of my review. I am however unable to raise my score because we have a disagreement on the novelty of the work. The ideas of low-rank matrix factorization and several other ad-hoc tweaks exist in the literature on compression of neural networks. While your work looks at the federated learning paradigm, and it utilizes an algorithm that helps learning take place in a federated setting, I maintain that these ideas by themselves are incremental in nature, when contrasted with the existing literature, and are not sufficient to warrant a publication in ICLR in the main conference. Perhaps this disagreement is because of I and you possibly coming from different communities, and I would let the AC and SAC resolve this disagreement.

---

### Decision · Program_Chairs · 2023-01-20

**Decision:**

Reject

**Justification For Why Not Higher Score:**

One of the reviewers had argued in favor of rejection (and I agree with the reviewer).

**Justification For Why Not Lower Score:**

-

**Metareview: Summary, Strengths And Weaknesses:**

The paper studies federated learning in scenarios where the clients do not have sufficient resources to train a large model and, additionally, the clients do not want to share any intermediate data and/or labels with the server. The authors propose a methodology, called Principal Sub-Model (PriSM), in which the convolution kernels are mapped to the so-called principal kernels given by the singular value decomposition (SVD) at the server and then clients are asked to update low-rank approximations of the convolution kernels that are given by a subset of the principal kernels.

The reviews had some major concerns, most of which were addressed through authors' responses, but some still remain. In particular, one of the reviewers has raised the issue of novelty, and some other reviewers believed that some more concrete analysis (e.g. theory) might be helpful to better demonstrate the main messages of the paper.

All in all, once the concerns of the reviews are addressed, I believe that the paper will be a great contribution to the field of FL.